# BENCHMARKING SAMPLE SELECTION STRATEGIES FOR BATCH REINFORCEMENT LEARNING

## ABSTRACT

Training sample selection techniques, such as prioritized experience replay (PER), have been recognized as of significant importance for online reinforcement learning algorithms. Efficient sample selection can help further improve the learning efficiency and the final performance. However, the impact of sample selection for batch reinforcement learning algorithms, where we aim to learn a near-optimal policy exclusively from the offline logged dataset, has not been well studied. In this work, we investigate the application of non-uniform sampling techniques in batch reinforcement learning. In particular, we compare six variants of PER based on various heuristic priority metrics that focus on different aspects of the offline learning setting. These metrics include temporal-difference error, n-step return, self-imitation learning objective, pseudo-count, uncertainty, and likelihood. Through extensive experiments on the standard batch RL datasets, we find that non-uniform sampling is also effective in batch RL settings. Furthermore, there is no single metric that works in all situations. Our findings also show that it is insufficient to avoid the bootstrapping error in batch reinforcement learning by only changing the sampling scheme.

## 1 INTRODUCTION

A key question in machine learning is to select the suitable training samples (Katharopoulos & Fleuret, 2018). Many prior works proved that an appropriate sample selection strategy, *i.e.*, removing redundant data or selecting samples according to their hardness, usually significantly improves the learning efficiency and final performance (Bengio et al., 2009; Schaul et al., 2015; Fan et al., 2017). Similarly, sample selection also plays a crucial role in reinforcement learning (RL) (De Bruin et al., 2018). A notable example is the sample selection problem for experience replay (ER) in off-policy RL (Fedus et al., 2020), where an agent reuses stored experiences from a buffer while interacting with the environment. For example, Prioritized Experience Replay (PER) (Schaul et al., 2015), which samples high error transitions more frequently, is now widely used in different *state-of-the-art* (SOTA) off-policy RL algorithms (Barth-Maron et al., 2018; Hessel et al., 2018; Kapturowski et al., 2018).

Batch RL, also known as offline RL, refers to the problem of learning a near-optimal policy from a fixed offline buffer (Lange et al., 2012). Due to the wide availability of logged-data and the increasing computing power, batch RL holds the promise for successful real-world applications (Levine et al., 2020). Especially for the scenarios where collecting online data is time-consuming, dangerous or unethical, *i.e.*, robotics, self-driving cars and medical treatments (Gulcehre et al., 2020). While most off-policy RL algorithms are applicable in the offline setting, they usually suffer from the bootstrapping error (Fujimoto et al., 2018b; Kumar et al., 2019) due to out-of-distribution (OOD) state-action pairs. Different solutions are proposed to mitigate this problem, *i.e.*, adding constraints (Fujimoto et al., 2018b; Wu et al., 2019), imitating behavior policy (Chen et al., 2019; Zolna et al., 2020), learning dynamics models (Yu et al., 2020; Kidambi et al., 2020; Argenson & Dulac-Arnold, 2020), incorporating uncertainties (Wu et al., 2021), learning ensembles (Agarwal et al., 2020), or learning pessimistic value functions (Kumar et al., 2020; Buckman et al., 2020; Jin et al., 2021).

Unlike the wide application of PER in online off-policy RL, the non-uniform sampling strategy is largely ignored in recent batch RL algorithms. Inspired by the success of PER (Schaul et al., 2015) in the online setting, one natural question to ask is that what is the counterpart of PER in batch RL?

This problem is appealing for several reasons: (1) In some real-world applications, the size of the offline dataset is usually increasing though we have no access to the real environment. For example, we would have ever-growing medical records from the hospitals (Raghu, 2019), or recorded videos from dash cams (Yu et al., 2018). (2) As the D4RL offline benchmark (Fu et al., 2020) shows that – for most existing methods, more samples do not necessarily lead to better performance. That is, a batch RL agent sometimes under-performs in a large combined buffer. Given the success achieved by PER in online RL, we are curious that if similar technique could help to develop more robust batch RL agents (Fujimoto et al., 2020).

Some prior works proposed different sample selection strategies in batch RL. For example, Optimal Sample Selection (OSS) (Rachelson et al., 2011) introduced a meta-learning algorithm which selects optimal samples according to a cross entropy search method for tree-based Fitted Q-Iteration (FQI) (Ernst et al., 2005) with a known dynamics model. Recently, Best-Action Imitation Learning (BAIL) (Chen et al., 2019) proposed to select high-performing samples with a learned value function in behavior cloning. Another related line of research is to reweight sampled transitions. For example, Advantage-Weighted Regression (AWR) (Peng et al., 2019) and Advantage-weighted Behavior Model (ABM) (Siegel et al., 2020) both used reward-weighted regression (Peters et al., 2010) to learn the policy. Further, Uncertainty Weighted Actor Critic (UWAC) (Wu et al., 2021) adopted a dropout-uncertainty estimation method (Gal & Ghahramani, 2016) and reweighted samples according to their estimated uncertainties. However, it is unclear which sample selection strategy is preferred in batch RL, thereby demanding more investigations.

To this end, in this work, we study the sample selection problem in batch RL (De Bruin et al., 2018). We follow the PER framework by assigning samples with different priorities (Schaul et al., 2015). Crudely, there are two types of metrics to evaluate sample importance. Firstly, we can design a heuristic metric based on our prior knowledge, *i.e.*, temporal-difference (TD) error. Secondly, we can use an end-to-end approach to learn a metric for each sample, for example, we can use off-policy evaluation (OPE) methods (Voloshin et al., 2019; Fu et al., 2021) to evaluate the goodness of current policy as the metric. However, existing OPE methods usually need to learn a model for each evaluation (Le et al., 2019), which makes the learning-based metric approach to be computationally expensive. Therefore, in this paper, we focus on the heuristic metric-based approach and leave the learning-based metric approach for future work. In particular, we benchmark six variants of PER based on different heuristic priority metrics in order to understand which sample selection strategy might be preferred in batch RL.

## 2 PRELIMINARIES

### 2.1 BATCH REINFORCEMENT LEARNING

We consider the standard Markov Decision Process (MDP) (Puterman, 2014) $\mathcal{M} = \langle \mathcal{S}, \mathcal{A}, T, r, \gamma \rangle$. $\mathcal{S}$ and $\mathcal{A}$ denote the state and action spaces. $T(s'|s, a)$ and $r(s, a)$ represent the dynamics and reward function, and $\gamma \in [0, 1)$ is the discount factor. A policy $\pi(a|s)$ defines a mapping from state to distributions over actions. The goal of an RL agent is to learn a policy $\pi(a|s)$ that maximizes the expected cumulative discounted reward $J(\pi) := \mathbb{E}_\pi \left[ \sum_{t=0}^\infty \gamma^t r_t \right]$. The performance of the policy can be defined by the Q-function $Q^\pi(s, a) := \mathbb{E}_\pi \left[ \sum_{t=0}^\infty \gamma^t r_t | s_0 = s, a_0 = a \right]$ and value function $V^\pi(s) := \mathbb{E}_\pi \left[ \sum_{t=0}^\infty \gamma^t r_t | s_0 = s \right]$, where $\mathbb{E}_\pi[\cdot]$ is the expected result when following the policy $\pi$. Once given the optimal Q-function $Q^*(s, a) = \arg\max_\pi Q^\pi(s, a)$, we can derive an optimal policy as $\pi^*(a|s) = \arg\max_a Q^*(s, a)$ (Sutton & Barto, 2018).

In (tabular) Q-learning, we solve for the $Q^*$ by iterating the Optimality Bellman Operator $\mathcal{T}^*$, defined as $\mathcal{T}^* Q(s, a) \leftarrow r + \gamma \max_{a'} Q(s', a)$ (Bertsekas & Tsitsiklis, 1996). To solve problems with large state space, we can use a parameterized Q-function $Q_\theta(s, a)$ to approximate $Q^*$. In practice, we optimize the parameters by a $\mu$-weighted L2 projection $\Pi_\mu(Q)$ (Fu et al., 2019), which minimizes the empirical Bellman error loss: $\Pi_\mu(Q) = \min_\theta \mathbb{E}_{(s,a,r,s') \sim \mu} \left[ (\mathcal{T}^* Q_\theta(s, a) - Q_\theta(s, a))^2 \right]$.

Batch RL, also known as offline RL, aims to learn a near-optimal policy from a fixed dataset (Lange et al., 2012) $\mathcal{D}$, representing a series of timestep tuples $(s_t, a_t, r_t, s_{t+1})$. Furthermore, the dataset can be collected by agents with different policies from different control tasks, including non-RL policies, such as human demonstrations (Levine et al., 2020). Some early works such as Fitted Q-Iteration (FQI) (Ernst et al., 2005) and Neural Fitted Q-Iteration (NFQ) (Riedmiller, 2005), which

formulate the original RL problem as a sequence of supervised regression problem, are shown to be sample efficient in solving various real-world problems (Pietquin et al., 2011; Cunha et al., 2015). On the other hand, some recent studies show that current deep off-policy RL algorithms usually fail in challenging batch RL problems due to bootstrapping error (Fujimoto et al., 2018b; Kumar et al., 2019). That is, the OOD action $a'$ might lead to unrecoverable over-estimation error through max operator in the Bellman backup. The over-estimation problem is particularly detrimental in the offline setting where the agent has no access to interact with the real environment to get the feedback to fix the estimation error (Kumar et al., 2020).

## 2.2 Non-uniform Sampling with Experience Replay

Experience replay (ER) (Lin, 1992) has been a de facto component for modern deep RL algorithms. By reusing previous collected experiences from the replay buffer, ER helps to reduce sample complexity and stabilize training in off-policy RL (Mnih et al., 2013; Lillicrap et al., 2015). For some real-world problems where collecting online data is expensive or time consuming, *i.e.*, robotics or self-driving cars, the ability to learn good policies from pre-collected data is crucial for successful real-world applications (Cabi et al., 2019).

A number of works (Schaul et al., 2015; Andrychowicz et al., 2017; Liu et al., 2019; Sun et al., 2020; Fujimoto et al., 2020) show that applying different non-uniform sampling strategies in ER can significantly improve the learning efficiency. Especially for problems where there are many redundant transitions (Schaul et al., 2015), or the reward signal is sparse (Andrychowicz et al., 2017). A notable example is the Prioritized Experience Replay (PER) (Schaul et al., 2015), where the probability of sampling a certain transition $(s_t, a_t, r_t, s_{t+1})$ is proportional to the absolute TD error. However, it is still an open question that which priority metric is optimal to value the importance of samples (De Bruin et al., 2018).

# 3 Related Work

## 3.1 Sample Selection with Experience Replay

Many prior works have sought to analyze the mechanism of experience replay, both empirically (De Bruin et al., 2018; Fedus et al., 2020) and theoretically (Fujimoto et al., 2020; Li et al., 2021). Similar to our work, (De Bruin et al., 2018) investigated a number of proxies, *i.e.*, age, TD error, and exploration noise, to decide which experience to store in the replay buffer and how to sample from the replay buffer. Likewise, (Fu et al., 2019) used a "unit-testing" framework to study Q-learning with function approximators and found that a sampling scheme with wider coverage improves performance. Further, (Fedus et al., 2020) conducted a systematic analysis of experience replay in $Q$-learning methods and provided two insights – (1) Increasing the buffer capacity is preferable, because it has a broader data coverage. (2) Decreasing the age of the oldest policy improves the performance, because it contains more high-quality on-policy data. While these insights help us to understand the mechanism of experience replay, they are less practical in the batch RL setting, where the given offline dataset is fixed (Lange et al., 2012).

A number of variants of ER have been introduced to further improve the learning efficiency (Schaul et al., 2015; Andrychowicz et al., 2017; Novati & Koumoutsakos, 2019; Liu et al., 2019; Sun et al., 2020). One of the most popular variants is the Prioritized Experience Replay (PER) (Schaul et al., 2015), which proposed to use the absolute TD error $|\delta(i)|$ as the priority metric and the probability of sampling the $i$-th transition is:

$$p(i) = \frac{p_i^\alpha}{\sum_j p_j^\alpha}, \quad p_i = |\delta(i)| + \epsilon \quad \text{or} \quad p_i = \frac{1}{\text{rank}(i)}, \tag{1}$$

where $\alpha$ is a hyper-parameter, $\epsilon$ is a small positive constant to avoid zero priority, priority $p_i$ is the value of $|\delta(i)|$ or the inverse rank of $|\delta(i)|$. In addition, Hindsight Experience Replay (HER) (Andrychowicz et al., 2017) proposed to re-label visited state as goal states to overcome hard exploration problems with sparse rewards. Competitive Experience Replay (CER) Liu et al. (2019) later introduced an automatic exploratory curriculum by formulating an exploration competition between two agents. On the other hand, Remember and Forget Experience Replay (ReF-ER) (Novati & Koumoutsakos, 2019) classified samples as "near-policy" and "far-policy" by the importance weight

$\rho = \pi(a|s)/\mu(a|s)$ between current policy $\pi$ and the behavior policy $\mu$, and compute gradients only with near-policy samples. Similarly, Attentive Experience Replay (AER) (Sun et al., 2020) selects samples according to the similarities between the transition state and current state.

Recently, Loss-Adjusted Prioritized (LAP) experience replay (Fujimoto et al., 2020) built the connection between the non-uniform sampling scheme in PER and loss functions. It shows that any loss function $\mathcal{L}_1$ evaluated with uniform sampling ($i \sim \mathcal{D}_1$) is equivalent to another loss function $\mathcal{L}_2$ that is evaluated with non-uniformly sampled data ($i \sim \mathcal{D}_2$):

$$\mathbb{E}_{i\sim\mathcal{D}_1}\left[\nabla_Q\mathcal{L}_1(\delta(i))\right] = \mathbb{E}_{i\sim\mathcal{D}_2}\left[\frac{p_{\mathcal{D}_1}(i)}{p_{\mathcal{D}_2}(i)}\nabla_Q\mathcal{L}_1(\delta(i))\right] = \mathbb{E}_{i\sim\mathcal{D}_2}\left[\nabla_Q\mathcal{L}_2(\delta(i)),\right] \tag{2}$$

where $\delta(i)$ is the TD error of the $i$-th sample and the two loss functions follows $\nabla_Q\mathcal{L}_2(\delta(i)) = \frac{p_{\mathcal{D}_1}(i)}{p_{\mathcal{D}_2}(i)}\nabla_Q\mathcal{L}_1(\delta(i))$. Moreover, Valuable Experience Replay (VER) (Li et al., 2021) proved that the absolute TD error $|\delta(i)|$ is an upper-bound of different value metrics of experiences in $Q$-learning.

## 3.2 SAMPLE SELECTION IN BATCH REINFORCEMENT LEARNING

A pioneering work that applied sample selection strategy in batch RL is the Optimal Sample Selection (OSS) method (Rachelson et al., 2011). More specifically, OSS is a model-based RL (MBRL) approach (Sutton & Barto, 2018) where a known dynamics model is available to generate Monte Carlo rollouts for policy evaluation. Moreover, OSS introduced a meta-learning algorithm to select optimal samples according to the cross entropy search method (Rubinstein & Kroese, 2004) for tree-based Fitted Q-Iteration (FQI) (Ernst et al., 2005). Recently, Best-Action Imitation Learning (BAIL) (Chen et al., 2019) proposed to learn a special value function $V_\phi(s)$, called upper envelope, that upper bounds the cumulative discounted return $G_i = \sum_{t=i}^{T}\gamma^{t-i}r_t$ from starting from state $s_i$ to the end of the episode (max horizon $T$). The learned upper envelope $V_\phi(S)$ is then used to filter high-quality samples to train a behavior cloning policy (Pomerleau, 1991).

Another related line of research is to reweight samples (Tirinzoni et al., 2018; Peng et al., 2019; Wu et al., 2021). Unlike previous methods that actively select samples from the buffer, these methods still adopt uniform sampling while assigning different weights to each sample to compute the loss function. For example, Advantage-Weighted Regression (AWR) (Peng et al., 2019) first formulated the RL problem as a supervised regression problem, and then used a learned value function to train the policy $\pi(a|s)$ via reward-weighted regression (Peters et al., 2010), which assigns higher weights to samples with large advantage values. Similarly, Advantage-weighted Behavior Model (ABM) (Siegel et al., 2020) adopted reward-weighted regression in policy training to focus more on good actions. On the other hand, Uncertainty Weighted Actor Critic (UWAC) (Wu et al., 2021) used Monte Carlo Dropout (Gal & Ghahramani, 2016) to approximate the epistemic uncertainty (Kendall & Gal, 2017) for samples in batch RL dataset. The goal of UWAC is to assign lower weights to samples with higher epistemic uncertainty in order to mitigate the bootstrapping error caused by OOD state-action pairs (Fujimoto et al., 2018b; Kumar et al., 2019).

## 4 METHODOLOGY

### 4.1 BACKBONE ALGORITHMS

In this work, we select TD3BC (Fujimoto & Gu, 2021) and PER (Schaul et al., 2015) as the backbone algorithms for benchmarking sample selection strategies in batch RL. TD3BC is a minimalist batch RL algorithm which simply adds a behavior cloning term to the TD3 algorithm (Fujimoto et al., 2018a). While being simple, TD3BC achieves comparable performance w.r.t. other SOTA batch RL algorithms (Kostrikov et al., 2021; Kumar et al., 2020) on the standard batch RL benchmark (Fu et al., 2020). Moreover, TD3BC is able to run significantly faster than previous methods by removing additional computations overheads.

In particular, TD3BC made two small modifications upon the origin TD3 algorithm (Fujimoto & Gu, 2021). Firstly, it adds a behavior cloning regularization term to the policy update in order to keep the learned policy $\pi$ to stay close to the behavior policy $\mu$:

$$\pi = \arg\max_\pi \mathbb{E}_{(s,a)\sim\mathcal{D}}\left[\lambda Q(s,\pi(s)) - (\pi(s) - a)^2\right], \tag{3}$$

where $\lambda$ is a parameter to trade-off RL and imitation. Secondly, it normalizes the states $s$ in the offline dataset by $s' = (s - s_\mu)/s_\sigma$ where $s_\mu$ and $s_\sigma$ are the mean and stander deviation.

In terms of the non-uniform sampling strategy, we follow the PER framework (Schaul et al., 2015) in which the probability to sample transition $i$ is $p(i) = p_i^\alpha / \sum_j p_j^\alpha$, where $p_i$ is the priority of transition $i$ and parameter $\alpha$ determines how much prioritization is used. In this paper, we investigate the problem of how the choice of priority metric matters in batch RL. We use both the proportional PER and rank-based PER in the experiment depending on the used priority metric. Proportional PER is a more popular baseline, while rank-based PER is more robust to outliers especially when different metrics have inconsistent scales.

## 4.2 PROPOSED METRICS

Here, we introduce six different priority metrics that we use in the experiment. We select these metrics based on prior insights of what data might be preferred in batch RL. Depending on whether the metric changes in the experiment, we could further divide them into static and dynamic metrics. A summary of these metrics is shown in Table 1.

Table 1: List of proposed metrics. Depending on if the value is fixed during training, we divide them into static and dynamic metrics. Some metrics are more computationally expensive, *i.e.*, requiring a dynamics model or behavior policy.

| Metric | Type | Motivation | Prioritization | Extra computation |
|--------|------|------------|----------------|-------------------|
| TD-Error | Dynamic | Reducing redundant samples | Proportional PER | - |
| N-step Return | Static | Selecting good samples | Rank PER | - |
| GSIL | Dynamic | Selecting good samples | Proportional PER | A second buffer |
| Pseudo-count | Static | Avoiding OOD samples | Rank PER | Hash table |
| Uncertainty | Static | Avoiding OOD samples | Rank PER | Probabilistic ensemble |
| Likelihood | Static | Being more on-policy | Rank PER | Behavior policy |

**TD error**. We first select the TD error as our primary baseline, and test how well does the naïve PER (Schaul et al., 2015) perform in the batch RL setting. We adopt the most popular proportional PER, and the priority for the $i$-th transition is $p_i = |\delta(i)| + \epsilon$, where $|\delta(i)|$ is the absolute TD error and $\epsilon$ is a small positive constant to avoid zero priority. The motivation of using TD error to select samples is that small absolute TD error samples contain less information for our model to learn from (Moore & Atkeson, 1993).

**N-step return**. Secondly, we select the n-step return as the proxy to evaluate the goodness of samples. Uncorrected n-step return has been shown to be an effective technique that greatly improves performances (Hessel et al., 2018; Fedus et al., 2020; Rowland et al., 2020). Similar to the idea of BAIL (Chen et al., 2019), we hypothesis that samples with higher n-step return is more likely to be high-quality samples. Unlike Monte Carlo return which requires a full trajectory, n-step return are much more practical in real-world problems, where we usually only have partial trajectories without a terminal state. Specifically, the n-step return for transition $i$ is $R_i^N = \sum_{t=0}^{N-1} \gamma^t r_{i+t}$, where $N$ is the horizon length. Notice that, the n-step return is fixed during the experiment which we only need to compute once in the data pre-processing step. Given the different reward scales across tasks, we use a rank based PER in the experiment $p_i = 1/\text{rank}(R_i^N)$.

**Generalized SIL**. Our third metric is inspired by the Self-Imitation Learning (SIL) (Oh et al., 2018), which exploits past good experiences. In particular, SIL imitates past good experiences by optimizing following actor-critic loss function:

$$\mathcal{L}_{value}^{sil} = \frac{1}{2} \| [R - V_\theta(s)]_+ \|^2, \quad \mathcal{L}_{policy}^{sil} = -\log \pi_\theta(a|s) [R - V_\theta(s)]_+ \tag{4}$$

where $R = \sum_{t=0}^{\infty} \gamma^t r_t$ is the cumulative discounted return starting from state $s$ after taking action $a$, and $[x]_+ = \max(0, x)$. The motivation of SIL is intuitive that policy $\pi_\theta(a|s)$ should imitate action

$a$ if it is high-performing, such that $R > V_\theta(s)$. Generalized Self-Imitation Learning (GSIL) (Tang, 2020) later extends the original SIL to deterministic actor-critic setting with n-step TD-learning. We follow GSIL to set the priority for the $i$-th transition to be $p_i = \left[R_i^N - Q_\theta(s_i, a_i)\right]_+ + \epsilon$, where $R_i^N$ is the n-step return and $\epsilon$ is a small positive constant.

**Pseudo-count**. Although batch RL do not concern the exploration problem (Osband et al., 2016). We can still borrow some insights from an exploration perspective to distinguish useful samples. For example, in the experiment, we test the efficacy of Pseudo-count (Ostrovski et al., 2017; Tang et al., 2017) for sample selection in batch RL. We follow the #Exploration (Tang et al., 2017) model to use locality-sensitive hashing (LSH) method, *i.e.*, SimHash (Charikar, 2002), to convert continuous state $s \in \mathbb{R}^D$ to discrete $k$-dimension hash codes:

$$\phi(s) = sgn(Ag(s)) \in \{-1, 1\}^k, \tag{5}$$

where $g(\cdot)$ is an optional preprocessing function and $A \in \mathbb{R}^{k \times D}$ is a random matrix. In the batch RL dataset, a small pseudo-count $N_i$ of a state-action pair $(s_i, a_i)$ means the dataset has insufficient coverage around these data. Hence, it would be more likely to suffer from the OOD sample problem when learning from these samples. In the experiment, we use a rank-based PER and set the priority of the $i$-th sample to be $p_i = 1/\text{rank}(N_i)$.

**Uncertainty**. Inspired by UWAC (Wu et al., 2021), we also attempt to use the epistemic uncertainty to evaluate the sample importance. UWAC adds a dropout layer before every weight layer (Gal & Ghahramani, 2016) and approximate uncertainty of state-action pair $(s, a)$ by the variance of predicted $Q(s, a)$. To exactly analyze how does the uncertainty-based sampling influence the performance, we adopt the probabilistic ensemble (Chua et al., 2018) method instead of change the origin TD3BC model. Specifically, we first train an ensemble of $M$ probabilistic dynamic models $\{T_1, T_2, \cdots, T_M\}$ (Pineda et al., 2021), where each dynamic model $T_i(s_{t+1}|s_t, a_t) = \mathcal{N}(\mu_{\theta_i}(s_t, a_t), \Sigma_{\theta_i}(s_t, a_t))$ outputs a Gaussian distribution with diagonal covariances parameterized by $\theta_i$. For the $i$-th transition $(s_i, a_i, r_i, s_{i+1})$, we approximate its epistemic uncertainty by the standard deviation of $\sigma_i = \text{std}(\{\mu_{\theta_1}(s_i, a_i), \cdots, \mu_{\theta_M}(s_i, a_i)\})$. We use a rank-based PER to assign higher priority to samples with smaller uncertainty, that is $p_i = 1/\text{rank}(\sigma_i^{-1})$.

**Likelihood**. The last metric we test in the experiment is the likelihood of the behavior model (Kostrikov et al., 2021). Similar to previous constrained based batch RL algorithms (Fujimoto et al., 2018b; Wu et al., 2019; Kumar et al., 2019), we want to make the learned policy $\pi(a|s)$ to stay close to the behavior policy $\mu(a|s)$. Therefore, we first learned a behavior policy with a mixture of Gaussian model (Kostrikov et al., 2021) and used the likelihood as the priority. We use a rank-based PER where $p_i = 1/\text{rank}(\log \mu(a_i|s_i))$ is the priority for the $i$-th sample.

## 5 EXPERIMENT

In this section, we compare different PER variants with the proposed metrics on a variety of batch RL continuous control tasks (Fu et al., 2020). We seek to address the following questions in the experiments: (1) Does non-uniform sampling scheme also help to improve the performance in batch RL? (2) Which priority metric is preferred in the batch RL setting?

**Datasets**. We evaluate different sample selection strategies on the widely-used D4RL gym Mujoco benchmark (Todorov et al., 2012; Fu et al., 2020), including three environments (halfcheetah, hopper, and walker2d) and five dataset types (random, medium, medium-replay, medium-expert, expert), yielding a total of 15 datasets. These datasets differ in many aspects, e.g., number of transitions, quality of behavior policy, and data coverage. We seek to validate the robustness of each sample selection strategy in different domains.

**Experiment setup**. For the backbone algorithm, we use the author-provided implementation for TD3BC [1]. We maintain two replay buffers for the GSIL metric as in the origin paper [2] (Tang, 2020), where the first buffer stores single-step transitions to train TD3 and the second buffer stores n-step transitions to compute the GSIL loss. In addition, we use the MBRL-Lib [3] (Pineda et al., 2021) to

---

[1] https://github.com/sfujim/TD3_BC

[2] https://github.com/robintyh1/nstep-sil

[3] https://github.com/facebookresearch/mbrl-lib

train the probabilistic ensemble, and implement the SimHash according to EPG [4] (Houthooft et al., 2018). Parameters for the PER are taken from the original paper (Schaul et al., 2015). We follow exactly the same experimental setup as (Fujimoto & Gu, 2021), in which we train for 1 million time steps and evaluate every 5000 time steps for 10 episodes. More details are in the Appendix.

**Results**. We report the final performance of different priority metrics in Table 2 and plot the learning curves in Figure 1. We make several observations: (1) Non-uniform sampling strategy is also effective in batch RL, for example, the most performant method in each environment is usually a non-uniform sampling strategy. (2) There is no single metric that is consistently the best performer. (3) In some environments, such as Hopper-Medium and Hopper-Expert, different sampling schemes perform very similar. In light of these results, we conclude that offline datasets are quite complicate and multiple factors can influence the sample priority. In environments with relatively low dimensions, such as Hopper, the learned policy is less affected by the sampling scheme.

Table 2: Performance of different priority metrics in the D4RL datasets. We report the average normalized score over the final 10 evaluations over 3 seeds ($\pm$ standard deviation).

| | | Uniform | TD-Error | Nstep-Return | GSIL | Pseudo-Count | Uncertainty | Likelihood |
|---|---|---|---|---|---|---|---|---|
| **Random** | HalfCheetah | $11.2 \pm 1.3$ | $11.1 \pm 1.1$ | $10.3 \pm 0.6$ | $9.1 \pm 2.0$ | $11.3 \pm 1.3$ | $\mathbf{11.4 \pm 1.2}$ | $11.0 \pm 0.6$ |
| | Hopper | $11.0 \pm 0.0$ | $10.9 \pm 0.1$ | $11.0 \pm 0.0$ | $10.9 \pm 0.0$ | $\mathbf{11.1 \pm 0.0}$ | $10.8 \pm 0.1$ | $11.0 \pm 0.0$ |
| | Walker2d | $0.9 \pm 0.6$ | $1.7 \pm 1.1$ | $2.6 \pm 0.8$ | $\mathbf{5.1 \pm 0.3}$ | $2.4 \pm 0.6$ | $2.3 \pm 1.7$ | $1.8 \pm 0.6$ |
| **Medium** | HalfCheetah | $42.9 \pm 0.1$ | $42.8 \pm 0.3$ | $\mathbf{43.9 \pm 0.5}$ | $43.2 \pm 0.2$ | $43.3 \pm 0.4$ | $42.9 \pm 0.4$ | $42.4 \pm 0.1$ |
| | Hopper | $\mathbf{99.9 \pm 0.1}$ | $99.6 \pm 0.4$ | $99.4 \pm 0.6$ | $99.8 \pm 0.1$ | $99.7 \pm 0.1$ | $99.8 \pm 0.1$ | $99.8 \pm 0.2$ |
| | Walker2d | $77.3 \pm 0.9$ | $78.2 \pm 1.0$ | $77.3 \pm 1.2$ | $77.9 \pm 1.3$ | $77.2 \pm 0.7$ | $76.9 \pm 0.6$ | $\mathbf{79.4 \pm 0.6}$ |
| **Medium Replay** | HalfCheetah | $43.1 \pm 0.4$ | $43.3 \pm 0.1$ | $\mathbf{43.5 \pm 0.5}$ | $42.8 \pm 0.2$ | $43.3 \pm 0.5$ | $43.4 \pm 0.2$ | $43.3 \pm 0.0$ |
| | Hopper | $\mathbf{32.1 \pm 1.3}$ | $30.3 \pm 0.8$ | $31.4 \pm 0.7$ | $30.6 \pm 1.9$ | $31.9 \pm 0.3$ | $31.1 \pm 1.8$ | $31.7 \pm 1.6$ |
| | Walker2d | $24.3 \pm 4.6$ | $23.8 \pm 2.5$ | $17.4 \pm 2.7$ | $15.2 \pm 9.4$ | $\mathbf{29.0 \pm 3.6}$ | $26.4 \pm 1.0$ | $24.8 \pm 1.1$ |
| **Medium Expert** | HalfCheetah | $92.4 \pm 1.5$ | $\mathbf{96.9 \pm 1.7}$ | $87.8 \pm 3.4$ | $96.5 \pm 2.1$ | $91.3 \pm 1.7$ | $88.0 \pm 3.3$ | $84.4 \pm 4.0$ |
| | Hopper | $112.0 \pm 0.1$ | $106.2 \pm 1.1$ | $110.7 \pm 1.7$ | $111.6 \pm 0.9$ | $\mathbf{112.2 \pm 0.0}$ | $109.8 \pm 2.2$ | $111.4 \pm 0.6$ |
| | Walker2d | $95.7 \pm 4.2$ | $96.9 \pm 3.0$ | $90.8 \pm 2.6$ | $103.1 \pm 4.1$ | $96.9 \pm 4.6$ | $\mathbf{103.9 \pm 2.1}$ | $81.4 \pm 23.6$ |
| **Expert** | HalfCheetah | $\mathbf{105.9 \pm 0.7}$ | $103.5 \pm 1.5$ | $102.4 \pm 1.6$ | $105.4 \pm 1.0$ | $104.0 \pm 1.2$ | $103.8 \pm 0.2$ | $104.5 \pm 0.9$ |
| | Hopper | $\mathbf{112.3 \pm 0.0}$ | $112.2 \pm 0.1$ | $112.2 \pm 0.1$ | $112.1 \pm 0.1$ | $112.2 \pm 0.1$ | $111.8 \pm 0.6$ | $44.6 \pm 47.9$ |
| | Walker2d | $105.0 \pm 2.0$ | $104.5 \pm 1.8$ | $105.5 \pm 1.7$ | $103.8 \pm 1.3$ | $104.2 \pm 1.1$ | $\mathbf{105.9 \pm 0.4}$ | $104.1 \pm 3.3$ |
| # Beat baseline | | - | 5 | 4 | 5 | **8** | 6 | 5 |

**Sampling scheme and bootstrapping error**. We also plot the learned $Q_1$ function in TD3 (Figure 2) to check for the bootstrapping error problem (Fujimoto et al., 2018b; Kumar et al., 2019). We can see that all the sampling schemes learn explosive $Q$ values in the Walker2d-Random environment, which implies that non-uniform sampling schemes fail to avoid the bootstrapping error. In addition, we can observe that the sampling scheme affects the learned $Q$ function, where the likelihood metric usually learns a relatively smaller $Q$ values and N-step return metric learns a relatively higher $Q$ values. This corresponds to the inductive bias of each metric. For example, a transition with higher N-step return is more likely to have large single-step return, which leads to higher $Q$ values in the Bellman backup. On the other hand, in the experiment, a transition with high likelihood is more likely to have lower reward. This may because most transitions in the dataset are at early stage of a trajectory, which have lower rewards. In addition, from figure 2, we can also observe that the bootstrapping error is not the only problem which prevents us to learn a good offline policy. For example, we did not suffer from severe bootstrapping error in HalfCheetah-Random and Hopper-Random environment, but we still fail to exploit the offline dataset to learn performant policies. This maybe the fixed behavior cloning parameter to be overly-constrained in such datasets with low data-quality. Thus, we believe that there is promise in further improving performance by relaxing the fixed behavior cloning parameter through a dynamic evaluation of current data-quality.

**Some problems of the proposed metrics**. Here, we summarize some shortcomings of the heuristic metric-based sample selection strategies. Firstly, we may need extra computations to compute the priority metrics (Table 1). For example, training the probability ensemble to estimate the sample

---

[4]https://github.com/openai/EPG

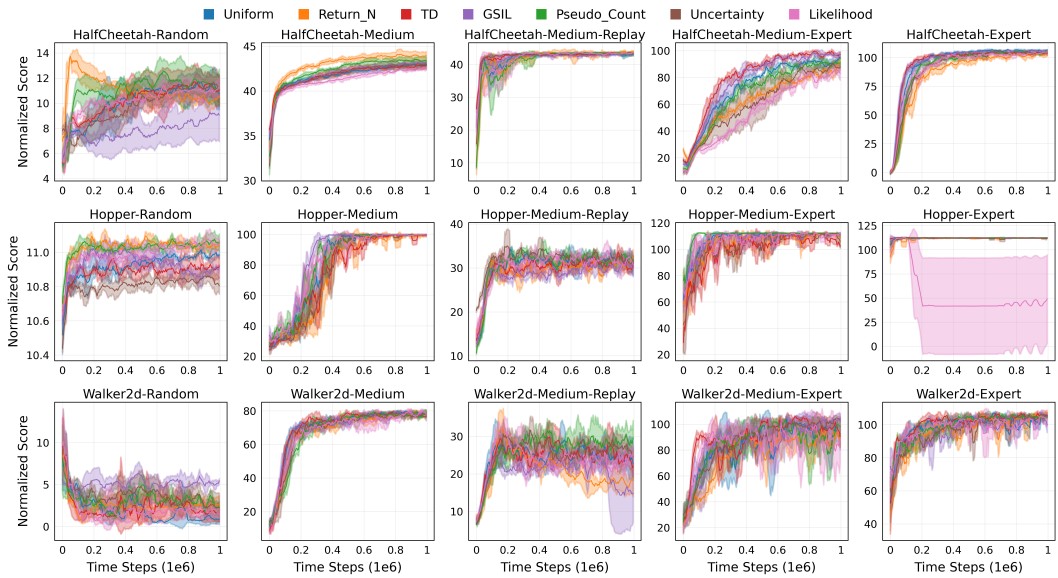

Figure 1: Results are averaged across 3 random seeds with shaded areas representing the standard deviation. We can observe that non-uniform sampling strategies is also effective in batch RL, and there is no priority metric that works in all situations.

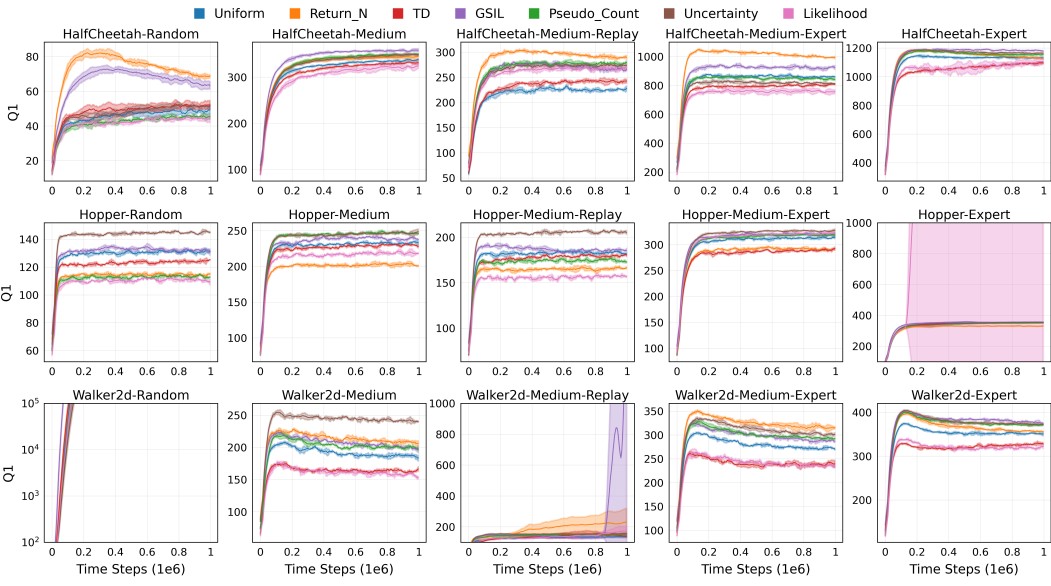

Figure 2: Learned $Q_1$ function in TD3. We can observe that non-uniform sampling schemes fail to avoid the bootstrapping error as in the Walker2d-Random environment.

uncertainty would be quite time-consuming. In addition, these extra models require further parameter tuning which may be problematic in the offline setting. Secondly, another critical problem of the metric-based sample selection method is that how exact is the metric for selecting a good sample. For example, a transition with low uncertainty or high likelihood is not necessarily a good sample for policy learning. Instead, it might be better to use these metrics as thresholds to filter bad samples. In particular, a low uncertainty transition may not be a good sample, but a high uncertainty transition is more likely to be a bad one. Thirdly, in the experiment, we compute the priority metric metric for transition $(s_i, a_i, r_i, s_{i+1})$ based the current state-action pair $(s_i, a_i)$. However, in the batch RL setting, the bootstrapping error comes from the OOD state-action pair $(s_{i+1}, a_{i+1})$. Therefore, it might be more effective to compute the priority metric based on the next state-action pair $(s_{i+1}, a_{i+1})$. We leave these shortcomings for future work.

## 6    CONCLUSION AND FUTURE WORK

In this paper, we perform empirical analysis on non-uniform sample selection strategies in batch reinforcement learning (RL). In particular, we compare different variants of Prioritized Experience Replay (PER) based on various heuristic sample priority metrics, including temporal-difference error, n-step return, self-imitation learning objective, pseudo-count, uncertainty and likelihood. Our experiments show that non-uniform sampling is also effective in the batch RL setting. However, there is no single priority metric that work in all situations, which shows that the offline datasets are quite complicate and multiple factors can influence the sample priority. A shortcoming of our work is that the proposed metric only focus on current state-action pairs and requires extra computations. A future direction is to learn a priority metric end-to-end with off-policy policy evaluation (OPE) methods. Another interesting future direction is to utilize unsupervised representation learning, *i.e.*, self-supervised learning, to extract useful hidden representations to assist the sample selection task in batch RL.

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

# A  APPENDIX

Here, we introduce some details of our experiments. For the PER model, we use the hyperparameters recommended in the origin paper (Schaul et al., 2015), and set $\alpha = 0.6$, $\beta = 0.4$. For the N-step return metric, we select the $N$ to be 20. For the GSIL metric, we follow the same experiment setup as in the origin GSIL paper (Tang, 2020). For the pseudo-count metric, we select the key dimension for Simhash by computing the 25% quantile and 50% quantile number (see Table 3). A small key dimension would lead to too many collisions while a large key dimension would lead to sparse collisions. We highlight the selected parameter for each environment we used in the experiment. For the uncertainty metric, we train an probabilistic ensemble with 7 models with early stopping. We use the default training parameters as in the MBRL-LIB (Pineda et al., 2021) package. For the likelihood metric, we use the official FBRC (Kostrikov et al., 2021) code to learn the behavior policy.

Table 3: Quantile number of the pseudo-count of each state-action pair in the offline dataset.

| Key Dimension | | 16 | | 24 | | 32 | | 48 | | 64 | | 128 | |
|---|---|---|---|---|---|---|---|---|---|---|---|---|---|
| | Quantile | 25% | 50% | 25% | 50% | 25% | 50% | 25% | 50% | 25% | 50% | 25% | 50% |
| Random | HalfCheetah | 29 | 85 | **1** | **3** | 1 | 1 | 1 | 1 | 1 | 1 | 1 | 1 |
| | Hopper | 1068 | 4377 | 321 | 1697 | 62 | 343 | **6** | **42** | 2 | 11 | 1 | 1 |
| | Walker2d | 56 | 201 | **2** | **10** | 1 | 3 | 1 | 1 | 1 | 1 | 1 | 1 |
| Medium | HalfCheetah | 670 | 3288 | 112 | 727 | 21 | 188 | **3** | **28** | 1 | 4 | 1 | 1 |
| | Hopper | 562 | 1931 | 125 | 626 | 41 | 208 | **7** | **43** | 2 | 13 | 1 | 1 |
| | Walker2d | 99 | 443 | 6 | 40 | **1** | **7** | 1 | 2 | 1 | 1 | 1 | 1 |
| Medium Replay | HalfCheetah | 7 | 29 | **1** | **3** | 1 | 1 | 1 | 1 | 1 | 1 | 1 | 1 |
| | Hopper | 57 | 205 | 8 | 37 | **2** | **10** | 1 | 1 | 1 | 1 | 1 | 1 |
| | Walker2d | 6 | 18 | **1** | **2** | 1 | 1 | 1 | 1 | 1 | 1 | 1 | 1 |
| Medium Expert | HalfCheetah | 838 | 4309 | 192 | 1241 | 23 | 210 | **4** | **43** | 1 | 8 | 1 | 1 |
| | Hopper | 405 | 1354 | 73 | 326 | 26 | 135 | **4** | **23** | 1 | 4 | 1 | 1 |
| | Walker2d | 218 | 1069 | 14 | 90 | **3** | **23** | 1 | 2 | 1 | 2 | 1 | 1 |
| Expert | HalfCheetah | 638 | 3831 | 149 | 947 | 23 | 198 | **2** | **12** | 1 | 2 | 1 | 1 |
| | Hopper | 1032 | 3683 | 175 | 718 | 25 | 123 | **4** | **24** | 2 | 8 | 1 | 1 |
| | Walker2d | 160 | 655 | 17 | 103 | **4** | **25** | 1 | 3 | 1 | 1 | 1 | 1 |

