# OpenReview forum: "Benchmarking Sample Selection Strategies for Batch Reinforcement Learning"
_ICLR.cc/2022/Conference — ICLR 2022 Submitted_

### Official Review · Reviewer_iELi · 2021-10-29

**Correctness:** 4
**Technical Novelty And Significance:** 1
**Empirical Novelty And Significance:** 3
**Recommendation:** 3
**Confidence:** 4

**Main Review:**

Pros:
* Overall the questions asked in this paper are worthwhile; I don't believe that PER has been applied to the offline setting before, so examining the efficacy of this is certainly an interesting direction.

Cons:
* This paper in its current state doesn't meet the standard for publication at a top venue as it lacks rigor for the following reasons:
   * Only one offline model-free RL algorithm was tested. More algorithms (e.g., CQL, MOPO, SAC etc.) should have been used. Perhaps using PER with a model-based method (and prioritising different hallucinated `(s,a,r,s')` tuples) could ameliorate the requirement of having a pessimistic MDP + reward penalty? Perhaps PER could be useful during model training?
   * Limited choice (6) of prioritization strategies; there should be more than this
   * Very low number of seeds (this should be at least 5)
   * No significance testing is done; are the blue highlighted results significant under a statistical test (e.g., Wilcoxon signed rank test)?
* The results themselves are not that compelling, and it doesn't appear that PER seems to help over uniform sampling much at all. I think a more directed piece of work that aims at creating a PER scheme that addresses distributional shift issues in offline RL, and show strong performance, would have been more suitable. Or perhaps work that seeks to explain why PER doesn't help that much, perhaps with some illustrative examples.
* The choice of citations is inappropriate at times. For instance, when model-based RL is introduced, the authors' cite recent work by Kaiser et al., but this is not a canonical nor an archetypal algorithm. Instead something like Dyna [1] or a review paper would be more appropriate. There are other examples of this littered throughout (e.g., citing Chua et al. for epistemic uncertainty), so the authors would do well to address this.

Nits:
* There are a few grammatical errors in this paper, for instance the use of 'pioneer' v.s. 'pioneering' at the top of section 3.2, and the repetition of 'metric' on page 9.

-------------------------------------

I have read the author's response but did not see much of what I requested in the updated manuscript (e.g., significance testing, additional algorithms, more compelling results, etc.). Therefore my rating stays the same.

Refs:

[1] "Integrated Architectures for Learning, Planning and Reacting based on Dynamic Programming", R. Sutton 1990.

**Summary Of The Paper:**

In this paper the authors' investigate the application of prioritized experience replay (PER) applied to offline/batch RL. They trial a variety of different priority metrics from the literature on the TD3+BC model-free algorithm and report results on the D4RL suite.

**Summary Of The Review:**

Overall this paper asks an interesting question, but fails in its execution for the reasons I have listed above. I believe it would be more suitable for publication at a workshop.

Based on this I recommend rejection.

---

> ### Author Response · Authors · 2021-11-22
> **Response to AnonReviewer4**
>
> Thank you for your detailed review!
>
> *> Lacks rigor. (only one offline model-free RL algorithm was tested; limit choice of prioritizationg strategies; low number of seeds; no significance testing is done)*
>
> - Thanks for the suggestion! We will try to add more experiments and the significance test.
>
>
> *> Results are not compelling.*
>
> - Thanks for the comment! We will try to explore a new PER scheme to address the distributional shift issue.
>
>
> *> Choice of citation, Typos.*
>
> - Thank you for pointing out. We will correct these in an updated version of the paper.

---

### Official Review · Reviewer_YFrP · 2021-11-01

**Correctness:** 3
**Technical Novelty And Significance:** 1
**Empirical Novelty And Significance:** 2
**Recommendation:** 3
**Confidence:** 4

**Main Review:**

Strength

- Table 1 (List of proposed metrics) is well-explained and clarifies the considered sampling variants.


Weakness

- As shown in the numerical results, sampling in offline RL may not be a critical issue compared with online learning. Unlike online learning, most of the samples in a fixed dataset may be visited.

- The proposed method lacks novelty. This paper does not propose any new algorithm for offline RL settings. Instead, this work combines TD3BC with the existing six variants of metric.

- While TD3BC has an advantage in its simplicity in implementation, other backbone algorithms for model-free offline RL should also be considered.


Question

- What will be the result if the following metric is used with proportional PER instead of rank PER: N-step Return, Uncertainty, and Likelihood?

**Summary Of The Paper:**

This paper investigated the effect of non-uniform sampling in an offline RL setting. Using TD3BC (Fujimoto and Gu, 2021) as a backbone offline RL algorithm, the authors applied prioritized experience replay (PER) to the sampling of TD3AC with variants of priority metric, including standard TD error, rank-based return, pseudo-count using a hash table, and the other three metrics. The authors insist that non-uniform sampling can be helpful in offline RL compared with usual uniform sampling. They also found that there is no one outperforming metric for prioritized sampling in offline RL settings.

**Summary Of The Review:**

Although this work raised an interesting question, this paper combines the existing algorithm and sampling metrics, which lack both novelty and empirical significance.


-------------------- Post Rebuttal ---------------------

Thanks for the response. I read the response carefully, and I maintain my original score. I hope the authors to further improve this work in the future.

---

> ### Author Response · Authors · 2021-11-22
> **Response to AnonReviewer3**
>
> Thank you for the comments!
>
> *> Sampling in offline RL may not be a critical issue compared with online learning.*
>
> - Non-uniform sampling strategy has been shown to be an effective technique to improve learning efficiency in online RL. However, there is few work that investigates which non-uniform sampling strategy is appealing in the offline setting. In this paper, we attempt to shed some light on this less investigated problem by benchmarking different sample priority metrics.
>
> *> Proposed method lacks novelty.*
>
> - Since the goal of this paper is to answer the question of -- how does non-uniform sampling strategies work in batch RL. Hence, we choose a minimalist offline model-free RL algorithm, and we pay more attentions to the design/selection of different sample metrics.
>
> *> Other backbone algorithms should be considered.*
> - Thanks for the suggestion! We will try to implement more backbone algorithms.
>
> *> What will be the result if the following metric is used with proportional PER instead of rank PER: N-step Return, Uncertainty, and Likelihood?*
> - The results would be less stable and requires more hyper-parameter tuning. Since each sample priority metrics have different scale, using the rank per allows us to avoid further hyper-parameter tuning, which is challenging in the offline learning setting, for each metric. In terms of the td-errors, we use the proportional PER because it is widely used in current model-free DRL algorithms.

---

### Official Review · Reviewer_8KRS · 2021-11-01

**Correctness:** 3
**Technical Novelty And Significance:** 2
**Empirical Novelty And Significance:** 2
**Recommendation:** 5
**Confidence:** 4

**Main Review:**

Strengths
1.	The paper is clearly written and easy to follow. It is self-contained and provides a good overview of existing literature.

Weaknesses
1.	I think the findings in this paper are not novel and significant. I guess the paper uses different sampling strategies to run TD3BC until convergence or for a fixed number of steps. In this case, isn’t the only difference between different sampling strategies the weighting on the loss function? In the online setting, different sampling strategies not only affect the learning at each time step but also the transitions the agent would see in the future. However, in the offline setting, the dataset is fixed, so it seems like the only impact is the weighting on the loss function? If this is the case, the paper basically shows that different weighting results in different performance and I don’t think that is a novel or significant finding.
2.	The empirical study is inconclusive. The paper mentions that the D4RL dataset is too complicated to draw any clear conclusion. Since the goal of the paper is not to propose a SOTA algorithm on “deep” RL environments, why not consider some smaller offline dataset or even discrete toy environments and see if any clear conclusion can be drawn? Moreover, in a small discrete environment, one might be able to show some relationship, e.g., using metric A performs well in environment B because it does C. I think that would be more informative than running on large complicated environments.

Questions:
1.  I am not sure what is the reason to use N-step return? It seems like N-step TD error might be a better choice since it matches the reasoning with PER. Also, what are the reasons for using Likelihood? It does not make sense to use only the state-action pairs closed to the behavior policy.
2. I am not sure what is the bootstrapping error in Section 5 and how it is connected to $Q_1$ in Figure 2?
3. The term “sampling selection” is confusing to me. Given a fixed dataset, even if we put a small sampling probability to a particular sample, we might still use the sample a sufficient number of times. I think it might be interesting to see a harder “sampling selection”: we drop a sample if the metric of the sample is below a threshold.



**Summary Of The Paper:**

The paper empirically investigates several sample selection strategies in offline RL based on TD3BC and the PER framework, including TD errors, N-step return, Generalized SIL, Pseudo-count, Uncertainty, and Likelihood. The paper finds that some sampling strategies improve the performance on D4RL dataset but they fail to avoid the bootstrapping error.

**Summary Of The Review:**

I am leaning towards a recommendation to reject the paper since the empirically findings in the paper are inconclusive and not significant as mentioned in the Main Review section.

---
After Rebuttal: Thank you for the response. My concerns are not addressed and I think the paper is not ready for publication. I hope the authors can keep improving the paper since it is an interesting topic. I will keep my original score and evaluation.

---

> ### Author Response · Authors · 2021-11-22
> **Response to AnonReviewer2**
>
> Thank you for the comments!
>
> *> Findings in this paper are not novel and significant.*
>
> - Sampling strategy indeed only affects the weighting on the loss function. In this paper, we want to answer the question of which kind of weighting works well in the batch RL setting. Due to the potential issue of extrapolation error, during the learning process, some weightings might lead to unrecoverable divergence. For example, a weighting with more weights w.r.t. samples with less data-coverage.
>
> *> Why not consider some smaller offline dataset or even discrete toy environments and see if any clear conclusion can be drawn?*
>
> - Thanks for the suggestion! We will try to run more experiments on some smaller (discrete) toy environments to find more conclusions.
>
> *> Why not consider some smaller offline dataset or even discrete toy environments and see if any clear conclusion can be drawn?*
>
> - Thanks for the suggestion! We will try to run more experiments on some smaller (discrete) toy environments to find more conclusions.
>
> *> Why use n-step return and likelihood as metric?*
>
> - As shown in BAIL (Chen et al., 2019), imitating samples with high value functions could learn performant policies in the batch RL setting. In BAIL, it uses the full trajectory return to select samples. However, in many practical problems, we do not have full trajectories in the offline dataset. Hence, we use the n-step return metric as a more practical proxy to select samples with high value functions. Further, the likelihood metric is used as a heuristic proxy to decrease the negative effects of learning from  out-of-distribution (OOD) samples.
>
> *> What is the bootstrapping error in Section 5 and how it is connected to Q1 in Figure 2?*
>
> - The bootstrapping (extrapolation) error (Fujimoto et al., 2018), refers to the over-estimation error caused by OOD action a' in the Ballman backup: $TQ(s, a) = r(s, a) +  max_{a'} Q(s', a')$. If (s', a') is not in the offline dataset, then the over-estimation error is not recoverable and will lead to a divergence of the value function. For example, the Q1 in Walker2d-Random in Figure 2 diverges no matter which priority metric we use.
>
> *> A harder “sampling selection”: we drop a sample if the metric of the sample is below a threshold.*
>
> - Thanks for the suggestion. We discussed the harder sample selection version in the future work section, and we will try to see how does the hard version work.
>
>
>
>
> [1] Chen, Xinyue, et al. "BAIL: Best-action imitation learning for batch deep reinforcement learning." arXiv preprint arXiv:1910.12179 (2019).

---

### Official Review · Reviewer_VhFt · 2021-11-03

**Correctness:** 4
**Technical Novelty And Significance:** 2
**Empirical Novelty And Significance:** 4
**Recommendation:** 3
**Confidence:** 3

**Main Review:**

**Strong points:**
- Sensible question: is a widely used method in online RL also useful in offline RL?
- While always inspired by online RL, I think at least some variants considered are not straightforward but rather an adaptation of an idea that works in online RL outside of PER. One example is using an idea that favors exploration in online RL to develop a sample weighting strategy.
- Empirical study are interesting imho.
- The paper offers interesting perspectives for further work.

**Weak points:**
- Imho, this paper assumes that the reader is extremely familiar with the state of the art in experience replay. I think this makes the paper perhaps harder than necessary to read. On the other hand, that type of paper must necessarily introduce a lot of algorithms quickly. I could not come up with a solution. Here are a few concrete examples:
   - Unless I missed it, some terms are never defined, such as behavior policy or temporal difference.
   - A lot of algorithms and variations are discussed and some are only introduced with a line of text.
- The empirical experiments are well done, but limited in scope. The paper considers only one algorithm and three environments.
- The paper requires significant additional proofreading. A few typos are listed below.
- I saw no mention that the code would be provided, although the paper strongly emphasizes the empirical validation.

**Questions:**
- When reading the beginning of the paper, I thought it was an empirical evaluation of existing online RL strategies in batch RL. However, it seems that some new methods are also considered, such as Pseudo-Count for example. I am surprised this is not highlighted better as a contribution of the paper.

**Typos:**
- end of page 3: is could be the value
- p5: samples with higher n-step return is
- p5: optimizing following actor-critic loss functions
- p5: motivation of SIL is intuitive that

**Summary Of The Paper:**

This paper empirically studies six variants of prioritized experience replay, typically used in online RL, in a batch RL setting. The comparison is performed using TD3BC on three D4RL Mujoco benchmark environment times 5 data sets. The experiments study the performance and bootstrapping errors. Among other things, it is shown that non-uniform sampling strategies are also interesting in a batch RL setting. The paper also discusses some shortcomings of these approaches and future directions.

**Summary Of The Review:**

Although mostly an empirical study, the research topic is interesting. I think the paper undersells a bit its contribution actually, as some variants considered are not trivial. It also provides interesting future directions. On the other hand, it is as bit hard to read, requires more proof-reading and the scale of the experiments is not big. Thus I believe it is a bit below the acceptance limit.

*After Rebuttal*: Thank you for your response. It seems that few changes have been made to the manuscript. I will keep my score unchanged. I wish the authors the best to improve the manuscript.

---

> ### Author Response · Authors · 2021-11-22
> **Response to AnonReviewer1**
>
> Thank you for your review!
>
> *> Some terms are never defined.*
>
> - Thanks for the suggestions. We've fixed them accordingly.
>
> *> The empirical experiments are limited in scope.*
>
> - In the experiment, we follows the experiment setting in the TD3-BC paper (Fujimoto et al., 2021). We will try to run more experiments.
>
> *> Related code.*
>
> - We will release the code later.
>
> *> New methods are not highlighted as a contribution.*
>
> - Thank you for pointing out. We will correct these in an updated version of the paper.
>
> [1] Fujimoto, Scott, and Shixiang Shane Gu. "A Minimalist Approach to Offline Reinforcement Learning." arXiv preprint arXiv:2106.06860 (2021).

---

### Decision · Program_Chairs · 2022-01-20

**Decision:**

Reject

**Comment:**

The paper empirically benchmarks multiple sample selection strategies for offline RL based on the prioritized experience replay framework, including TD errors, N-step return, Generalized SIL, Pseudo-count, Uncertainty, and Likelihood. These are all benchmarked for the base algorithm TD3BC. The experiments study the performance and bootstrapping errors. Among other things, it is shown that non-uniform sampling strategies are also interesting in a batch RL setting. The authors show that non-uniform sampling can be helpful in offline RL compared to uniform sampling but they fail to avoid bootstrap error. They also found that there is no one outperforming metric for prioritized sampling in offline RL settings.

The reviewers are in agreement that the question studied is a sensible and interesting one - Are PER strategies which are effective in online RL also useful for batch RL? The overall study conducted by the paper is clear and well presented.

While the study/benchmark and the results presented is clear, the reviewers point out the following shortcomings
1. The study is not comprehensive for this work to become a definitive exploration of this space of ideas. Only algorithm has been tested with these ideas.
2. The results of the study are unfortunately inconclusive - while there are benefits these are achieved via different strategies and as mentioned by the paper no clear conclusions can be drawn.

Since the paper is targeted purely as a benchmark, the originality aspect of the paper is naturally low. For benchmark papers in that case the impact factor squarely falls on comprehensiveness of the study and the emergence of some clear conclusions to further research in that area. The reviewers unanimously believe the paper falls short in both respects and therefore the decision.

Hopefully the authors can consider the feedback provided and incorporate it to improve the paper.